# A high-quality hourly, daily and monthly solar irradiance dataset in China during 1981-2014 based on MERRA-2 Reanalysis products

Wenmin Qin[1], Lunche Wang[1]*, Ming Zhang[1], Lan Feng[1], Hejin Fang[1], Hong Cai[1], Yulong Zhong[1], Qiqi Zhu[1], Chao Yang[2]

[1]Hubei Key Laboratory of Critical Zone Evolution, School of Geography and Information Engineering, China University of Geosciences, Wuhan, 430074, China;

[2] Institute of Geodesy and Geophysics, Chinese Academy of Sciences, Wuhan, 430077, China;

*Correspondence to*: Lunche Wang (wang@cug.edu.cn)

**Abstract.** Solar irradiance (SI) is the main driving factor contributing to climate change and energy balance between the land and atmosphere. High-quality records of global solar irradiance (GHI), direct normal irradiance (DNI) and diffuse solar irradiance (DIF) are of vital importance for solar applications, but the solar radiation observations are sparse around the world. As an alternative, numerous SI reanalysis data in grid format have been developed in regional and global scales. Among them, the MERRA-2 (Modern-Era Retrospective Analysis for Research and Applications, version 2) products could provide high quality SI records with acceptable accuracy and long temporal ranges. This study attempted to improve the accuracy of GHI records derived from MERRA-2 products, and to generate grid DNI and DIF datasets for all-sky conditions over mainland China during 1981-2014, based on the REST2 model and cloud transmittance estimates combining sunshine observations. The results indicate that the estimated GHI values (GHInew) show higher agreements with GHI measurement at 17 CMA (China meteorological administrations) stations than that for the GHI records derived from MERRA-2 products (MERRA-2 GHI). Then, grid GHI, DNI and DIF datasets (0.50° (lat) *0.625° (lon)) throughout China were constructed. The results indicated that the MERRA-2 GHI records may overestimate the GHI values over mainland China. Generally, the GHI and DNI values gradually decreased during 1981-2014, however, DIF values gradually increased from 1981 to 2014, especially in 1992 (DIF = 90.914 $Wm^{-2}$, anomaly DIF value = 15.544 $Wm^{-2}$). The Qinghai Tibetan Plateau has always been an area with the highest GHI, the highest DNI and the lowest DIF values, whereas the Sichuan Basin has always been an area with the lowest GHI, the lowest DNI and the highest DIF values. The grid GHI, DNI and DIF dataset generated in this study can assist in numerous solar studies and applications. We provide these solar





irradiance data in publicly available repository: https://doi.org/10.6084/m9.figshare.10026563 (Qin, W.
et al., 2019).
**Keywords:** global horizontal solar radiation; direct normal irradiance; diffuse horizontal solar radiation;
MERRA-2; China
**1 Introduction**
Solar energy is a clean, renewable and sustainable energy source for solar energy applications such
as photovoltaic energy utilization (Besharat et al. 2013; Purohit and Purohit 2015). China has the largest
thermal power generation of any country around the world, making it the largest emitter of greenhouse
gases (Amadei et al. 2013). The large demand for electricity and energy consumption has caused the
Chinese government to vigorously develop the concentrated solar thermal (CST) CSP industry (Li et al.
2014). China is also in the leading position regarding the construction and planned installed capacity of
CSP power generation around the world (Zhao et al. 2017). Therefore, accurate measurement of solar
irradiance (SI) is the basis and prerequisite for effective utilization of solar radiation resources (Qin et al.

43   2019).

Many observation networks have been constructed for providing SI records (in point format) in
China. The Baseline Surface Radiation Network (BSRN, Zhang et al. 2015), the World Radiation Data
Center (WRDC, Zhang et al. 2017), and the Global Energy Balance Archive (GEBA, Wild et al. 2017)
can provide SI records covering more than 2,000 observation stations around the world. In China,
according to the statistics of China Meteorological Data Network, there have been 122 solar radiation
measurement stations installed since 1957. In the 1990s, there were only 96 stations for solar radiation
measurement throughout China (Zou et al. 2017). However, because of the high cost of the site
construction and observation instruments, especially in remote areas with poor natural conditions, these
SI observation networks are still too scarce to support solar energy research and applications in China
(Qin et al. 2018).
In contrast with solar radiation observation stations, there are thousands of meteorological stations
covering mainland China. Thus, many studies have been conducted to construct GHI, DNI and DIF
datasets (in point format) in China using meteorological measurements at CMA stations. Tang et al. (2018)
first constructed a direct solar radiation data set in China with acceptable accuracy and high point density



(2474 CMA stations). Chen et al. (2019) applied five artificial intelligence models and one broadband
model for estimating direct solar radiation. The direct solar radiation results could be converted to DNI
by multiplying by the solar zenith angle. Feng et al. (2018) evaluated 15 empirical models for predicting
DIF values at 17 CMA radiation stations. However, the drawback to these solar radiation estimations is
that there are few DNI estimation results in Western China, especially in the Qinghai Tibetan Plateau due
to the extremely sparse meteorological measurements in Western China. Thus, SI records in grid format
covering mainland China with high spatial and temporal resolutions are urgently needed for solar
research and solar energy applications in China.
Numerous SI products in grid format have been created providing grid GHI, DNI and DIF records
with high spatial and temporal continuity covering mainland China; for example, the Global Energy and
Water Exchanges-Surface Radiation Budget Project (GEWEX-SRB, Raschke et al., 2006)) and the
International Satellite Cloud Climatology Project-flux data (ISCCP-FD, Lohmann et al., 2006)) can
provide solar radiation records throughout China with spatial resolution of $1°×1°$. SI can also be derived
from the GEDEX (Greenhouse Effect Detection Experiment) products developed by the NCAS British
Atmospheric Data Centre (NCAS BADC) (Sinha and Shine 1995). The Climate Data Record (CDR)
generated by NOAA can provide GHI records in China with long temporal ranges (1882-2019)
(Coddington et al. 2016). However, these products still cannot meet the requirements of solar energy
research in China needing GHI, DNI and DIF records with high accuracy and spatial resolution (Qin et
al. 2015).
Remote sensing is an alternative method to obtain GHI, DNI and DIF values in China with high
spatial resolution. GHI, DNI and DIF records could be derived from HelioClim (Blanc et al. 2011), MSG,
Meteosat (Möser and Raschke 1984), GOES (Gautier et al. 1980), MODIS (Qin et al. 2011), Himawari
(Bessho et al. 2016), and CM-SAF SARAH (Riihelä et al. 2015) observations. However, the accuracy of
these GHI, DNI and DIF records needs to be improved. Shi et al. (2018) evaluated the accuracy of the
estimated GHI values derived from the Advanced Himawari Imager (AHI) aboard Himawari-8 at 36
CERN (Chinese Ecosystem Research Network) stations. The results show that the GHI estimations did
not show good agreement with GHI measurements. Thus, many scientists have developed efficient
algorithms to improve the quality of solar radiation estimations in China using satellite images. Wei et
al. Wei et al. (2019) compared the accuracy of the estimated GHI values over China based on four
different AI models using AVHRR data, and they analyzed the spatial and temporal variations in GHI
over mainland China. Qin et al. (2015) developed an efficient physical parameterization (EPP) for
estimating GHI values using MODIS land and atmospheric products and evaluated the EPP model at 91
CMA stations in China. However, the spatial resolution ($1° \times 1°$) and spatial continuity of the estimation
results by the EPP model could not meet the requirements of solar energy research, which requires SI
records with high spatial resolution. Tang et al. (2016a) further improved the EPP model (EPP-TANG) by
combining the MODIS and MTSAT products in China. The spatial resolution of GHI estimations have
been improved to a 5 km spacing, but the spatial continuity of GHI estimations still restrict the
applicability of the EPP-TANG model in China. Liang et al. (2006) developed an efficient model based
on the look-up table method and the atmospheric radiation transfer model for incident GHI using MODIS
products. Zhang et al. (2014) further generated GHI, DNI and DIF products called GLASS (Global Land
Surface Satellite) covering China. Nevertheless, EPP, EPP-TANG and GLASS could only generate
instantaneous solar radiation values, thus they could not provide accurate daily GHI, DNI and DIF
records over mainland China (Tang et al. 2016b).

Reanalysis data is an alternative SI data source with acceptable accuracy and high spatiotemporal

continuity covering mainland China (Rienecker et al. 2011). ERA5 is the fifth generation of ECMWF
atmospheric reanalysis global climate data providing hourly and daily surface downward solar radiation
records from 1979 to present (Babar et al. 2019). SI values could also be derived from NCEP-DOE
AMIP-II reanalysis (Kanamitsu et al. 2002). The CRU JRA V2.0 dataset is also a data source with an
hourly downward solar radiation flux (Beck et al. 2017). The Climate Forecast System Reanalysis (CFSR)
developed by the National Oceanic and Atmospheric Administration (NOAA) could provide solar
radiation records from 1979 to present (Fuka et al. 2014). Using the GEOS-5 atmospheric general
circulation model (AGCM), the Modern-Era Retrospective Analysis for Research and Application
(MERRA) was stimulated by the National Aeronautics and Space Administration (NASA) Global
Modeling and Assimilation Office (GMAO), which could provide hourly, daily and monthly GHI records
during 1980-2019 at global scales (Bosilovich et al. 2011). The GHI values derived from MERRA-2
products were demonstrated to have good agreement with GHI measurements (Hodges et al. 2011;
Kennedy et al. 2011). Thus, the updated version (MERRA-2) was developed with numerous
improvements (Randles et al. 2017). In this study, it was supposed that the accuracy of GHI records in





MERRA-2 could be improved by integrating the effects of cloud transmittances. Moreover, the DNI and
DIF records are missing in previous MERRA-2 reanalysis data.
In what follows, GHI, DNI and DIF measurements at Wuhan station, Xianghe station, 40 CERN
stations and 17 first-class CMA meteorological stations throughout mainland China are used to evaluate
the performance of the estimated solar irradiance (GHI, DNI and DIF) values and GHI records derived
from MERRA-2 products during 1993–2014. In a subsequent step, the GHI, DNI and DIF databases
throughout mainland China are constructed using MERRA-2 products and sunshine duration
measurements at 2474 CMA stations. Finally, the spatiotemporal variations and possible influencing
factors on GHI, DNI and DIF over different climate zones and terrains in mainland China are investigated.
Overall, this study should prove helpful in solar resource and energy applications that need long-term
grid GHI, DNI and DIF data with moderate spatiotemporal resolution and acceptable accuracy. We
provide these solar irradiance data in publicly available repository (Qin, W. et al., 2019).
**2 Materials and methods**
**2.1  Sites and data processing**
Hourly GHI, DNI and DIF measurements at Xianghe station (BSRN) in China were used for
calculated the cloud transmittances for surface global horizontal solar radiation (GHI), direct normal
irradiance (DNI) and diffuse horizontal solar radiation (DIF). Hourly GHI measurements from 40 CERN
stations,hourly DNI and DIF measurements at Wuhan station (in Wuhan university), and Daily GHI,
DNI and DIF measurements during 1993-2014 at 17 CMA stations in China were used for evaluating the
model accuracy of the estimated hourly, daily and month GHI, DNI and DIF values generated in this
study. Meanwhile, the sunshine duration measurements during 1981-2014 that were routinely measured
at 2474 CMA stations over mainland China were also used to calculate the cloud transmittance of the
GHI, DNI and DIF. These meteorological data have been checked for data quality using various control
methods.

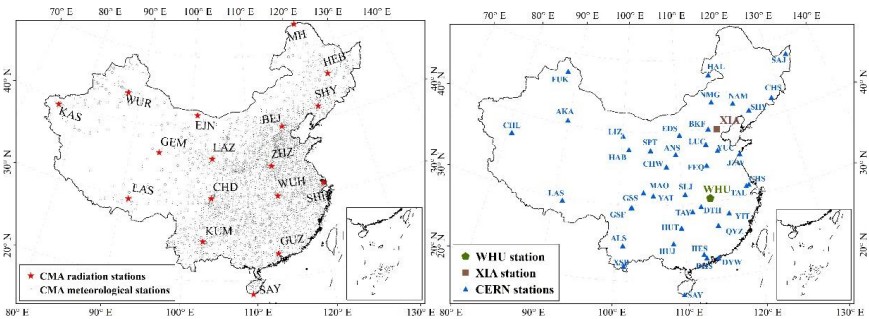

Figure 1. Spatial distributions of the CMA stations that are used in this study.

Figure 1 show the spatial distributions of the Wuhan station, Xianghe station and CMA meteorological stations that were used in this study. These stations covered most areas of China with distinct climatic and terrain features.

**2.2 REST2 Model**

REST2 is a physically based model for predicting hourly and daily broadband GHI, DNI and DIF values in clear sky conditions, which was first developed by (Gueymard 2003), then corrected and modified by (Gueymard 2012) (REST to REST2). REST2 has been validated as one of the best broadband solar radiation estimation models and has been widely used in numerous solar radiation research (Gueymard 2003). The REST2 model has corrected the diffusion calculation under low-AOD, near-Rayleigh conditions in the model. The GHI, DNI and DIF values in REST2 can be obtained using following equations:

$$\mathrm{DNI} = \tau_R \tau_g \tau_o \tau_n \tau_w \tau_a \mathrm{E}_0 \tag{1}$$

$$\mathrm{DIF} = 0.5 \tau_g \tau_o \tau_n \tau_w (1 - \tau_a \tau_R) \mathrm{E}_0 \tag{2}$$

$$\mathrm{GHI} = \mathrm{DNI} * \cos(\theta) + \mathrm{DIF} \tag{3}$$

where $\tau_R, \tau_g, \tau_o, \tau_n, \tau_w$ and $\tau_a$ are the transmittances for Rayleigh scattering, uniformly mixed gases absorption, ozone absorption, nitrogen dioxide absorption, water vapor absorption and aerosol extinction, respectively. $\mathrm{E}_0$ is the extraterrestrial solar radiation. $\theta$ is the solar zenith angle. These transmittances have been obtained accurately by fitting a large number of parametric runs of the SMARTS code to computationally efficient polynomial ratios (Gueymard 2012).



Considering the data availability, the hourly reanalysis meteorological records derived from the
MERRA-2 dataset during 1981-2014, including aerosol optical depth in band 550 (*AOD550)*, regional
ground albedo *(rog)*, air pressure (*p*) and precipitable water vapor (*w*) were used as model inputs for
REST2. The spatial resolution of the MERRA-2 dataset that was used in this study is $0.50°$ (lat) $*0.625°$ (lon).
More detailed descriptions and resulting equations of the REST2 model can be found in Ref (Gueymard

2008, 2012).

**2.3 Anusplin**
The sunshine duration measurements during 1981-2014 at 2474 CMA stations throughout mainland
China were used to calculate the cloud transmittances of GHI, DNI and DIF values. However, these CMA
stations are still too sparse to support the solar radiation estimations in this study. Therefore, using the
sunshine durations measurements at CMA stations, grid sunshine duration data (0.50° (lat) *0.625° (lon))
during 1981-2014 over mainland China were generated based on the Anusplin tool. The ANUSPLIN
package provides a facility for transparent analysis and interpolation of noisy multivariate data using
thin-plate smoothing splines, comprehensive statistical analyses, data diagnostics and spatially
distributed standard errors (Xu and Hutchinson 2013). The flowchart in the Anusplin tool was shown in
Figure 2. A detailed description of the Anusplin tool could be found in Ref (Hutchinson and Xu 2004).

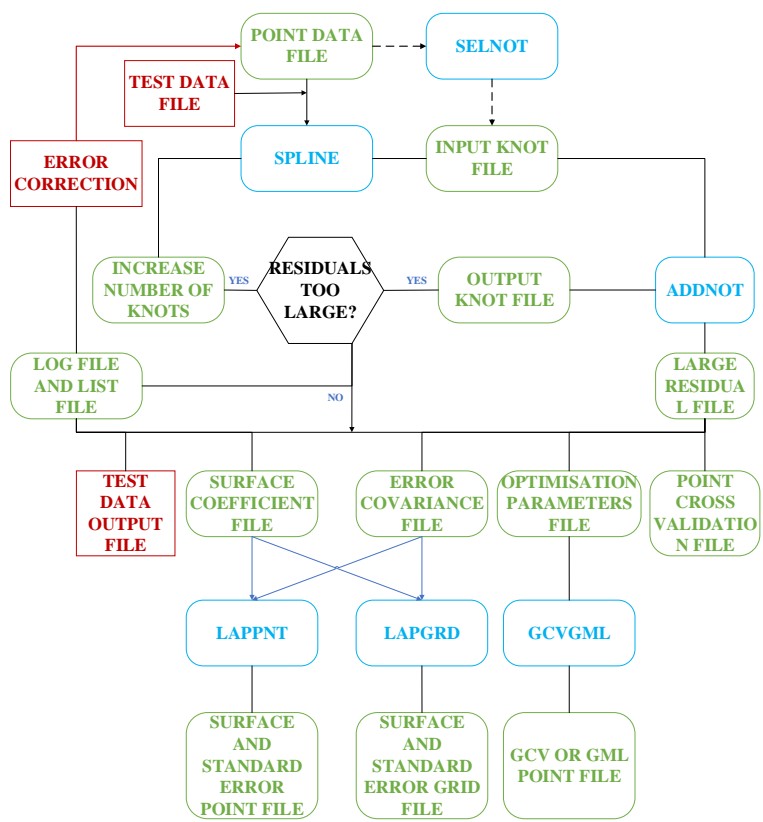


Figure 2. The flowchart of the processes of Anusplin tool.

**2.4 Comparisons of measures of fit**

In this study, 16 indicators were used to evaluate the model accuracy (Gueymard 2014). $N$ and bar,

respectively, indicate the number of data and mean of the variables; $e_i$ and $o_i$ are the modeled and
observed GHI, DNI and DIF values. These indicators are divided into four classes: Class A-indicators of
dispersion, Class B-indicators of overall performance, Class C-indicators of distribution similitude and
Class D-a global performance indicator.
**2.4.1 Class A – indicators of dispersion**

The Class A indicators are the root mean square error (RMSE), the mean absolute bias error (MAE),

the relatively root mean square error (RMSD), the relatively mean absolute bias error (MAD), the
correlation coefficient R, the standard deviation (SD), the slope of best-fit line (SBF), the uncertainty at
95% (U95), and the t-statistic (TS), which can be expressed as:





$$RMSE = \sqrt{\frac{1}{N}\sum_{i=1}^{N}(e_i - o_i)^2} \tag{4}$$

$$MAE = \frac{1}{N}\sum_{i=1}^{n}|e_i - o_i| \tag{5}$$

$$RMSD = \frac{100}{\overline{o_i}} \times \sqrt{\frac{1}{N}\sum_{i=1}^{N}(e_i - o_i)^2} \tag{6}$$

$$MAD = \frac{100}{\overline{o_i}} \times \frac{1}{N}\sum_{i=1}^{n}|e_i - o_i| \tag{7}$$

$$R = \frac{\sum_{i=1}^{n}(e_i - \overline{e})(o_i - \overline{o})}{\sqrt{\sum_{i=1}^{n}(e_i - \overline{e})^2}\sqrt{\sum_{i=1}^{n}(o_i - \overline{o})^2}} \tag{8}$$

$$SD = \frac{100}{\overline{o}} \times \left(\sqrt{\sum_{i=1}^{N}N(e_i - \overline{e})^2 - \sum_{i=1}^{N}(o_i - \overline{o})^2}\right)/N \tag{9}$$

$$SBF = \frac{\sum_{i=1}^{n}(e_i - \overline{e})(o_i - \overline{o})}{\sum_{i=1}^{n}(o_i - \overline{o})^2} \tag{10}$$

$$U95 = 1.96\sqrt{(SD^2 + RMSD^2)} \tag{11}$$

$$TS = \sqrt{(N-1)MBD^2/(RMSD^2 - MBD^2)} \tag{12}$$

### 2.4.2 Class B – indicators of overall performance


The Class B indicators are the Nash–Sutcliffe's efficiency (NSE), the Willmotts's index of agreement
(WIA) and the Legates's coefficient of efficiency (LCE), which can be expressed as:

$$NSE = 1 - \sum_{i=1}^{N}(e_i - o_i)^2 / \sum_{i=1}^{N}(o_i - \overline{o})^2 \tag{13}$$

$$WIA = 1 - \sum_{i=1}^{N}(e_i - o_i)^2 / \sum_{i=1}^{N}(|e_i - o_i| + |o_i - \overline{o}|)^2 \tag{14}$$

$$ICE = 1 - \sum_{i=1}^{N}|e_i - o_i| / \sum_{i=1}^{N}|o_i - \overline{o}| \tag{15}$$

### 2.4.3 Class C – indicators of distribution similitude


The Class C indicators are the Kolmo-gorov–Smirnovtest Integral (KSI), the relative frequency of
exceedance (OVER) and combined performance index (CPI), which can be expressed as:

$$KSI = \frac{100}{A_c}\int_{x_{min}}^{x_{max}}D_n dx \tag{16}$$

$$A_c = D_c(X_{max} - X_{min}) \tag{17}$$

$$D_c = \emptyset(N)/\sqrt{N} \tag{18}$$

$$OVER = \frac{100}{A_c}\int_{x_{min}}^{x_{max}}Max(D_n - D_c, 0)dx \tag{19}$$

$$CPI = (KSI + OVER + 2RMSE)/4 \tag{20}$$

where $D_n$ is the absolute difference between the two normalized distributions within irradiance interval
n, $X_{min}$ and $X_{max}$ are the minimum and maximum values of the binned reduced irradiance, x, and $A_c$ is a
characteristic quantity of the distribution. Detail descriptions of KSI, OVER and CPI indicators could be





found in Ref (Gueymard 2014).

**2.4.4 Class D –a global performance indicator (GPI)**

Although 16 indicators are introduced to reveal the model accuracy of GHI, DNI and DIF values,
too many indicators cannot reflect the overall accuracy of the estimated GHI, DNI and DIF values. In
this study, a global performance indicator (GPI) is used to represent the global performance of the
estimated GHI, DNI and DIF values (Despotovic et al. 2015). The GPI can be described by following
equation:

$$GPI_i = \sum_{j=1}^{n} a_j(\tilde{y} - y_{ij}) \tag{21}$$

where $\tilde{y}$ is the median of the scaled values of indicator $j$, $y_{ij}$ is the scaled value of indicator $j$ for model $i$,
and n is the number (16) of indicators. $a_j$ equals -1 for R, SBF, NSE, WIA and LCE, and equals 1 for
other indicators. The greater the accuracy of the model, the higher the value of the GPI.

**3 Result**

**3.1 Cloud transmittance for surface solar irradiance**

Due to the shape, type and phase variability in clouds, they have been considered to be the most
uncertain factor in estimating SI. In this study, the relative sunshine duration, defined as the ratio between
the measured sunshine duration and the maximum possible sunshine duration, (N) was introduced to
correct the cloud effect on hourly GHI, DNI and DIF values. Following the example of the Ångström-
Prescott equation, we parameterized the cloud transmittance (τc) as a function of the relative sunshine
duration (n/N), and the formula form was a quadratic polynomial formulation as follows:

$$\tau_c = \frac{R}{R_{clr}} = a + b\left(\frac{n}{N}\right) + c\left(\frac{n}{N}\right)^2 \tag{22}$$

where R is the hourly and daily all-sky GHI, DNI and DIF; $R_{clr}$ is the hourly and daily clear-sky GHI,
DNI and DIF; and n and N are the sunshine duration and the maximum possible sunshine duration,
respectively. The calibrated cloud transmittance for hourly GHI/DNI/DIF values are shown as the
following equations:

$$\tau_{c1} = 0.368 + 0.628\left(\frac{n}{N}\right) - 0.005\left(\frac{n}{N}\right)^2 \tag{23}$$

$$\tau_{c2} = 0.035 + 0.331\left(\frac{n}{N}\right) + 0.298\left(\frac{n}{N}\right)^2 \tag{24}$$

$$\tau_{c3} = 0.752 + 2.396\left(\frac{n}{N}\right) - 2.029\left(\frac{n}{N}\right)^2 \tag{25}$$

where $\tau_{c1}$, $\tau_{c2}$ and $\tau_{c3}$ are the cloud transmittance formula for hourly GHI, DNI and DIF values,



respectively.
The calibrated cloud transmittance for daily GHI/DNI/DIF values are shown as the following
equations:

$$\tau_{c1} = 0.280 + 0.954\left(\frac{n}{N}\right) - 0.299(\frac{n}{N})^2 \qquad (26)$$
$$\tau_{c2} = 0.024 + 0.227\left(\frac{n}{N}\right) + 0.619(\frac{n}{N})^2 \qquad (27)$$
$$\tau_{c3} = 0.959 + 4.115\left(\frac{n}{N}\right) - 4.232(\frac{n}{N})^2 \qquad (28)$$

where $\tau_{c1}, \tau_{c2}$ and $\tau_{c3}$ are the cloud transmittance formula for daily GHI, DNI and DIF values,
respectively.
**3.2 Validation of the estimated GHI, DNI and DIF at CMA stations**
Hourly GHI, DNI and DIF measurements at Wuhan stations, Xianghe stations and 40 CERN stations
were used to validate the accuracy of the estimated hourly GHI, DNI and DIF values. Daily GHI, DNI
and DIF measurements during 1993-2014 at 17 CMA meteorological stations are used for evaluating the
model accuracy of the daily estimated GHI, DNI and DIF values. The GHI records derived from
MERRA-2 products are also compared with the estimated GHI values in this study.



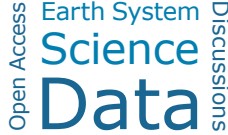

Table 1. Validation results of the estimated hourly mean GHI values at Xianghe and Wuhan station

| Stations | Solar irradiance without cloud transmittances | | | | | | Solar irradiance with cloud transmittances | | | | | |
| | WHU | | | XIA | | | WHU | | | XIA | | |
| Value | GHI | DNI | DIF | GHI | DNI | DIF | GHI | DNI | DIF | GHI | DNI | DIF |
| --- | --- | --- | --- | --- | --- | --- | --- | --- | --- | --- | --- | --- |
| RMSE | 280.52 | 329.55 | 208.81 | 140.90 | 283.13 | 70.46 | 129.30 | 177.60 | 197.13 | 57.64 | 125.54 | 58.17 |
| MAE | 204.56 | 288.19 | 170.70 | 64.34 | 162.46 | 33.04 | 97.33 | 133.82 | 163.57 | 27.11 | 55.20 | 28.14 |
| RMSD | 56.45 | 68.38 | 113.05 | 62.80 | 107.12 | 94.12 | 41.16 | 83.00 | 91.89 | 36.22 | 85.27 | 64.01 |
| MAD | 41.16 | 59.79 | 92.42 | 28.67 | 61.47 | 44.13 | 30.98 | 62.53 | 76.25 | 17.04 | 37.49 | 30.96 |
| MBD | -32.08 | -59.60 | -22.68 | -28.34 | -60.50 | 18.57 | 7.43 | -9.01 | -33.43 | 1.05 | -29.09 | -2.32 |
| SD | 56.44 | 68.37 | 113.04 | 62.80 | 107.12 | 94.11 | 41.15 | 82.99 | 91.88 | 36.22 | 85.26 | 64.01 |
| R | 0.62 | 0.63 | 0.08 | 0.91 | 0.69 | 0.85 | 0.88 | 0.47 | 0.36 | 0.97 | 0.86 | 0.89 |
| U95 | 156.46 | 189.52 | 313.34 | 174.07 | 296.92 | 260.87 | 114.08 | 230.05 | 254.69 | 100.40 | 236.34 | 177.42 |
| TS | 69.18 | 178.14 | 20.51 | 81.26 | 109.97 | 32.34 | 18.39 | 10.93 | 39.11 | 4.64 | 58.32 | 5.83 |
| NSE | -0.08 | -1.53 | -4.79 | 0.77 | 0.22 | 0.51 | 0.61 | 0.20 | -2.39 | 0.93 | 0.70 | 0.78 |
| WIA | 0.90 | 0.61 | 0.97 | 0.98 | 0.91 | 0.99 | 1.00 | 0.99 | 0.93 | 1.00 | 0.99 | 1.00 |
| LCE | 0.11 | -0.68 | -1.46 | 0.75 | 0.43 | 0.60 | 0.43 | 0.21 | -0.92 | 0.85 | 0.70 | 0.72 |
| KSI | 225.49 | 519.13 | 403.10 | 124.69 | 273.35 | 85.14 | 147.75 | 288.85 | 380.42 | 56.41 | 112.71 | 72.69 |
| OVER | 216.55 | 507.73 | 390.96 | 106.48 | 254.71 | 46.70 | 118.96 | 282.48 | 361.80 | 24.11 | 82.18 | 51.84 |
| CPI | 138.73 | 290.90 | 255.04 | 89.19 | 185.57 | 80.02 | 87.26 | 184.33 | 231.50 | 38.24 | 91.36 | 63.14 |
| OM | 496.96 | 481.98 | 184.71 | 224.37 | 264.30 | 74.87 | 314.18 | 213.99 | 214.53 | 159.12 | 147.23 | 90.88 |
| PM | 337.53 | 194.71 | 142.82 | 160.78 | 104.40 | 88.77 | 337.53 | 194.71 | 142.82 | 160.78 | 104.40 | 88.77 |
| GPI | 0.003 | -4.658 | -5.468 | 2.452 | -2.042 | 1.185 | 2.876 | -0.923 | -3.260 | 4.872 | 1.694 | 3.269 |







Table 1 illustrate the statistical indicators representing the model accuracy of the estimated hourly
GHI, DNI and DIF values at Wuhan and Xianghe station. The result indicated that the estimated hourly
GHI, DNI and DIF values show high agreements with the hourly GHI, DNI and DIF measurements. The
cloud has obvious effect on the accuracy of the estimated solar irradiance values. The modeling accuracy
have been significantly improved after incorporating the cloud transmittances for solar irradiance. The
GPI scores for GHI$_{WHU}$, DNI$_{WHU}$, DIF$_{WHU}$, GHI$_{XIA}$, DNI$_{XIA}$ and DIF$_{XIA}$ without cloud transmittances are
0.003, -4.658, -5.468, 2.452, -2.042 and 1.185, respectively; the GPI scores for GHI$_{WHU}$, DNI$_{WHU}$,
DIF$_{WHU}$, GHI$_{XIA}$, DNI$_{XIA}$ and DIF$_{XIA}$ with cloud transmittances are 2.876, -0.923, -3.260, 4.872, 1.694
and 3.269, respectively. Table S1 show the validation results of the estimated GHI values in different
CERN stations over mainland China. The estimated hourly GHI values show good agreements with the
hourly GHI measurements, but with distinct spatial variations over mainland China. Relatively large
model deviations are found in mountain and desert zones, due to the dramatic diurnal variations of the
climate factors (water vapor, temperature, cloud and pressure etc.) there, for example the GPI scores for
GSF, AKA, ALS, LAS and CHL are -9.869, -5.528, -2.685, -2.363 and -2.220, respectively.

Table S1. Validation results of the estimated hourly GHI values at CERN stations.

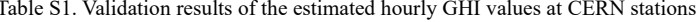

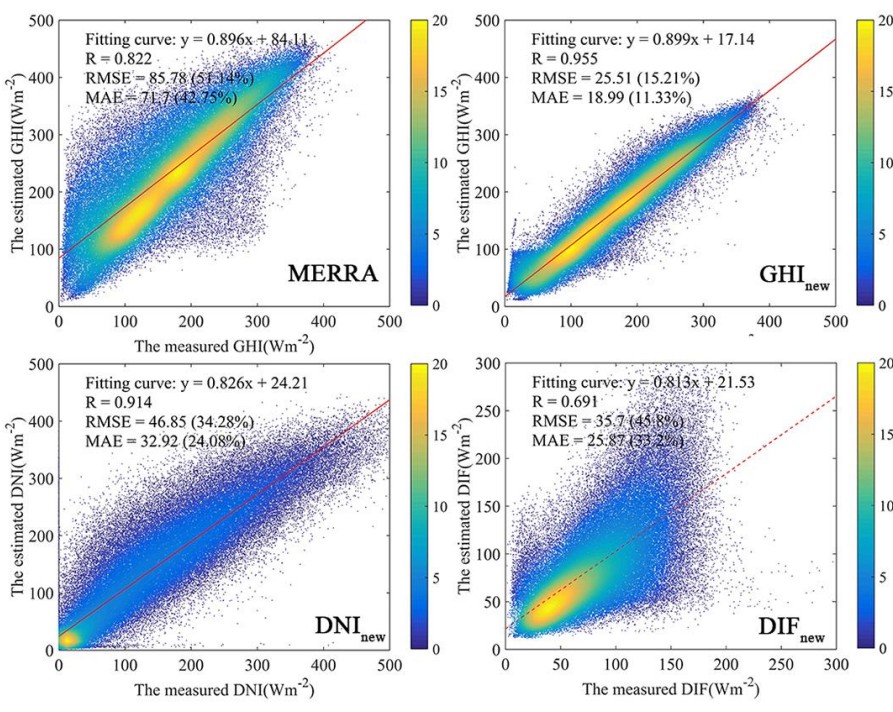


Figure 3. Validation of the daily mean GHI, DNI and DIF values at CMA stations.





Figure 3 is the scatter plot showing the model accuracy of MERRA-2 GHI records and the estimated
GHI, DNI and DIF values from the REST model. Table 2 specifies the statistical indicators representing
the model accuracy of the estimated GHI, DNI and DIF values. It is clear that the GHI estimations by
the REST model (GHInew) show greater agreement with the measured GHI values than with the
MERRA-2 GHI records. The RMSE, MAE, RMSD, MAD, MBD, SD, U95, TS, KSI, OVER and CPI
(Group 1 indicators) for MERRA-2 GHI records are significantly larger than for GHInew, while the R,
NSE, WIA, ICE (Group 2 indicators) for MERRA-2 GHI records are significantly lower than for GHInew.
The RMSE, MAE and R for MERRA-2 GHI records are 85.775 $Wm^{-2}$, 71.696 $Wm^{-2}$ and 0.822,
respectively. The RMSE, MAE and R for GHInew are 25.505 $Wm^{-2}$, 18.994 $Wm^{-2}$ and 0.955, respectively.
The accuracy of GHI records is significantly improved. The DNI and DIF estimations by the REST model
(DNInew and DIFnew) also show a high correlation with the ground DNI measurements. The RMSE,
MAE and R for DNInew are 46.853 $Wm^{-2}$, 32.917 $Wm^{-2}$ and 0.914, respectively. The RMSE, MAE and
R for DIFnew are 35.700 $Wm^{-2}$, 25.870 $Wm^{-2}$ and 0.690, respectively.
Table 2. The statistical indicators representing the model accuracy of the estimated daily GHI, DNI and
DIF values.

| Indicators | GHInew | DNInew | DIFnew | MERRA-2 GHI |
|---|---|---|---|---|
| RMSE | 25.52 | 46.85 | 35.70 | 85.81 |
| MAE | 19.01 | 32.91 | 25.87 | 71.72 |
| RMSD | 15.18 | 34.17 | 42.06 | 36.60 |
| MAD | 11.30 | 24.00 | 30.49 | 30.59 |
| MBD | -0.17 | -0.32 | -8.16 | -28.41 |
| SD | 15.18 | 34.17 | 42.06 | 36.60 |
| R | 0.95 | 0.91 | 0.69 | 0.82 |
| SBF | 1.01 | 1.01 | 0.59 | 0.75 |
| U95 | 42.07 | 94.71 | 116.59 | 101.44 |
| TS | 3.91 | 3.32 | 70.75 | 440.33 |
| NSE | 0.90 | 0.80 | 0.43 | 0.16 |
| WIA | 1.00 | 1.00 | 0.99 | 0.84 |
| LCE | 0.72 | 0.63 | 0.30 | 0.08 |
| KSI | 122.04 | 263.13 | 284.45 | 738.80 |
| OVER | 103.56 | 255.81 | 275.86 | 732.45 |
| CPI | 63.99 | 146.82 | 161.11 | 386.11 |

* The units for RMSE and MAE are $Wm^{-2}$; the units for RMSD, MAD, MBD, SD, U95, TS, KSI, OVER
and CPI are %; R, NSE, WIA, ICE are dimensionless indexes.

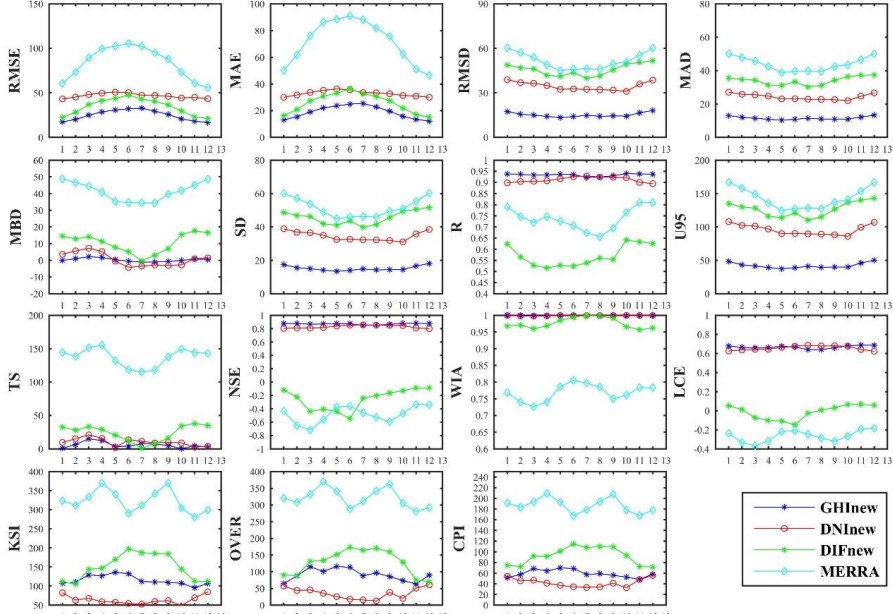


Figure 4. The statistical indicators representing the model accuracy of the estimated GHI, DNI and DIF
values in different months (The units for RMSE and MAE is MJ m-2day-1; the units for RMSD, MAD,
MBD, SD, U95, TS, KSI, OVER and CPI are %; R, NSE, WIA, ICE are dimensionless indexes.).

Figure 4 shows the statistical indicators representing the model accuracy of the estimated GHI, DNI

and DIF values in different months. The MERRA-2 GHI records are not as accurate as GHInew in all
months throughout the year. The values of Group 1 indicators for MERRA-2 GHI records are
significantly larger than GHInew, while the values of Group 2 indicators for MERRA-2 GHI records are
significantly lower than GHInew. The fluctuations in the values of Group 1 and Group 2 indicators for
MERRA-2 GHI records in Figure 4 are also more obvious than that for GHInew, DNInew and DIFnew
values, which further verified that the accuracy and robustness of GHI records are significantly improved
in this study.

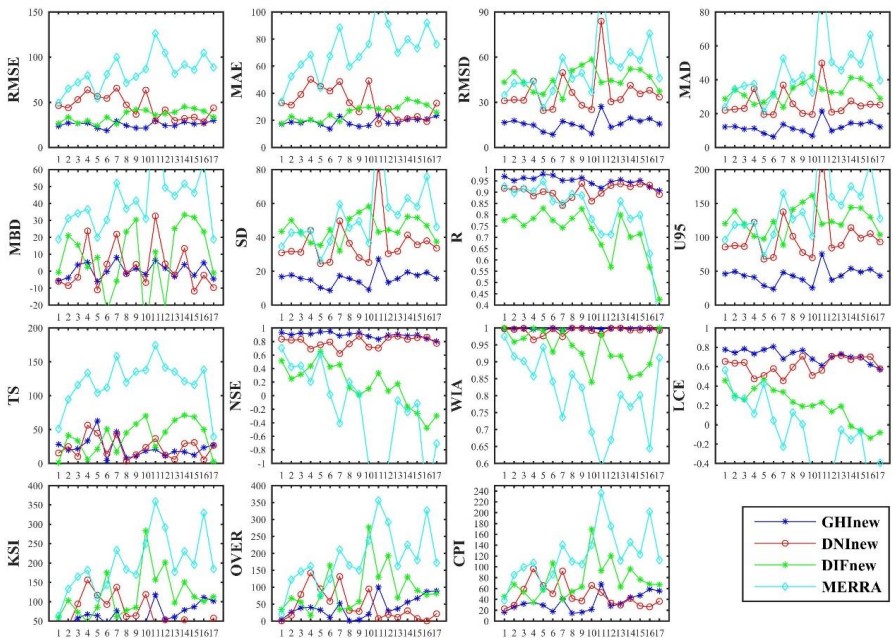

Figure 5. The statistical indicators representing the model accuracy of the estimated GHI, DNI and DIF values at 17 CMA stations (The number in x axis correspond to the ID in Table 1.).

The model accuracy of the MERRA-2 GHI records and the estimated GHI, DNI and DIF values are closely correlated to local climate and terrain features. Figure 5 illustrated the statistical indicators representing the model accuracy of the estimated GHI, DNI and DIF values at 17 CMA stations. It was clear that the GHInew performance was superior to MERRA-2 GHI records with higher accuracy and robustness. The values of Group 1 indicators for GHInew were significantly lower than for MERRA-2 GHI products in all months throughout the year, while the values of Group 2 indicators of GHInew were significantly higher than those of MERRA-2 GHI products in all months throughout the year.

Furthermore, the GPI scores for the MERRA-2 GHI and GHInew values are also calculated to show the overall model estimation error at 17 CMA stations over mainland China. Figure 6 show the spatial distribution of GPI scores for MERRA-2 GHI and GHInew values at 17 CMA stations in China. The accuracy of GHInew values is obviously higher than MERRA-2 GHI records with higher GPI scores. The mean GPI scores for GHInew and MERRA-2 are 3.079 and -3.079, respectively. Relatively larger estimation errors are found in the Sichuan Basin, which may be due to the strong atmospheric radiation dumping processes there (frequent rainy and cloudy weather). The GPI scores for MERRA-2 GHI and GHInew at the Chengdu station were 1.911 and -10.329, respectively. The accuracy of MERRA-2 GHI

and GHInew values were higher in arid zones and plateau zones with high GPI scores due to the relative
clear sky conditions there. The GPI scores for MERRA-2 GHI and GHInew at the Ejinaqi station were
3.284 and 0.753, respectively. The GPI scores for MERRA-2 GHI and GHInew at the Germu station
were 3.904 and -1.469, respectively.

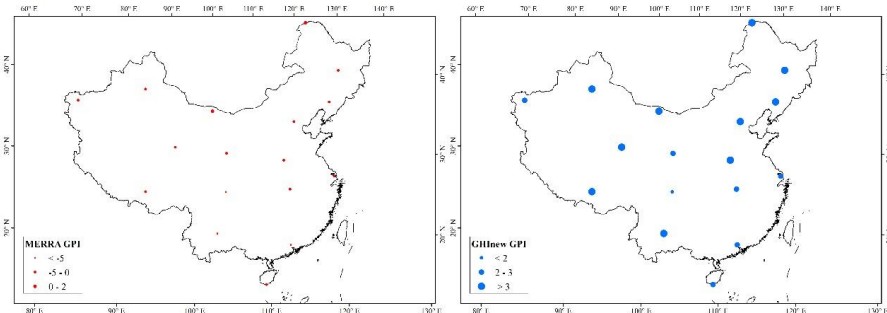


Figure 6. The GPI scores of the MERRA-2 GHI and GHInew values at 17 CMA stations. (The GPI is a
dimensionless index.).
Figure 7 shows the validation results of the monthly mean MERRA-2 GHI, GHInew, DNInew and
DIFnew values. It was obvious that the monthly mean GHInew, DNInew and DIFnew estimation results
could meet the requirement of the potential solar energy estimations and the proper installations of solar
power plants using CST with acceptable accuracy. The RMSE, MAE and R for the monthly mean
GHInew estimations were 14.745 $Wm^{-2}$, 10.602 $Wm^{-2}$ and 0.973, respectively. The RMSE, MAE and R
for the monthly mean DNInew estimations were 27.778 $Wm^{-2}$, 20.463 $Wm^{-2}$ and 0.922, respectively. The
RMSE, MAE and R for the monthly mean DIFnew estimations were 22.730 $Wm^{-2}$, 17.690 $Wm^{-2}$ and
0.798, respectively.

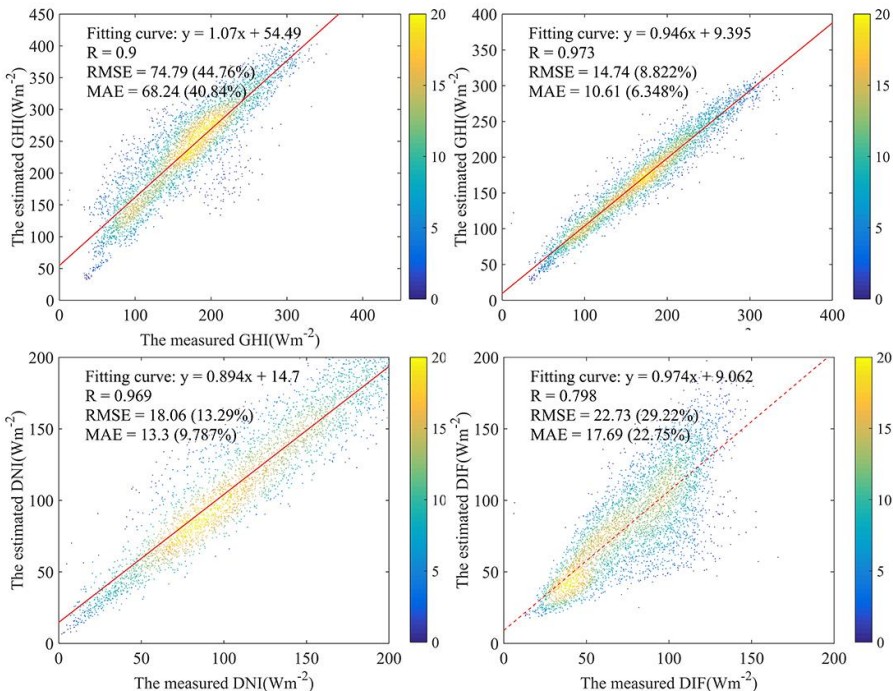

Figure 7. Validation of the monthly mean GHI, DNI and DIF values at CMA stations.

Overall, the MERRA-2 GHI products have been significantly improved in this study. Moreover, the

DNI and DIF datasets during 1981-2014 were generated with acceptable accuracy, which can be used
for solar energy research and applications.
**3.3 Spatial and temporal variations in surface solar radiation**

By applying the REST model, grid GHI, DNI and DIF datasets (0.50° (lat) *0.625° (lon)) during

1981-2014 throughout China are constructed. Figure 8 illustrates the spatial distributions of mean daily
GHI, DNI and DIF values from 1981-2014 in China. The MERRA-2 GHI records may overestimate the
GHI values in China, especially in the Sichuan Basin and Yungui Plateau, which may be due to ignoring
the effect of sunshine duration and clouds. The ranges of GHI values for MERRA-2 are 133.831 Wm$^{-2}$-
280.856 Wm$^{-2}$, while the ranges for GHI values by the REST models are 108.819 Wm$^{-2}$-246.134 Wm$^{-2}$.
The DNI values are closely correlated to the GHI values with similar spatial distribution patterns.
Generally, both GHI and DNI gradually decline from Northwestern China to Southeastern China.
However, DIF show a distinct spatial distribution pattern from that of GHI and DNI. The DIF values are
higher in Southeastern China and the Tarim Basin in Xinjiang Province. The Qinghai Tibetan Plateau is
always an area with the highest GHI, the highest DNI values and the lowest DIF values owing to the
relatively weak radiation dumping effect there, while the Sichuan Basin is always an area with the highest
DIF values and lowest GHI; the lowest DNI values are affected by the strong cloud cover effect.

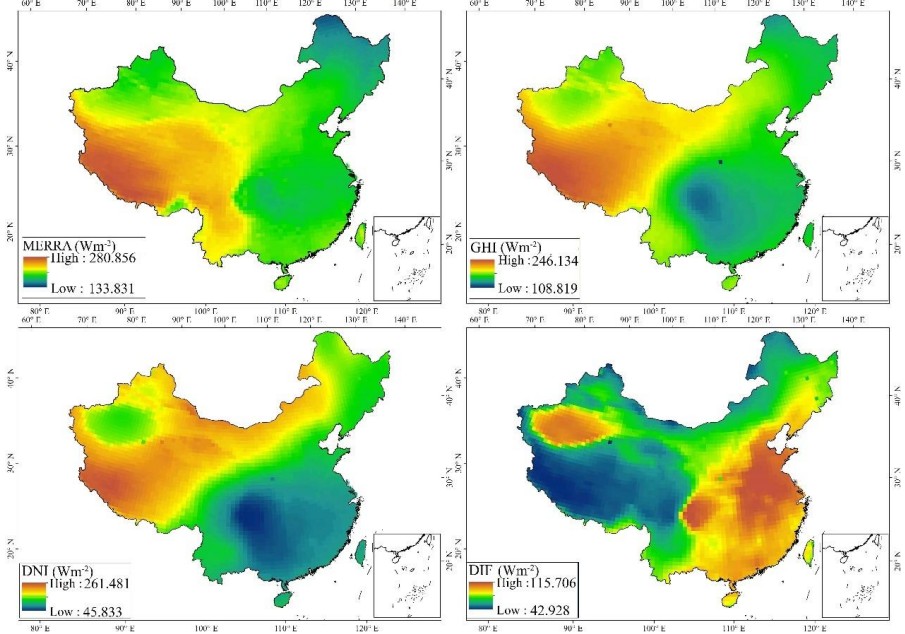

Figure 8. The spatial variation of GHI, DNI and DIF over mainland China (The units for GHI, DNI and
DIF are Wm⁻²).

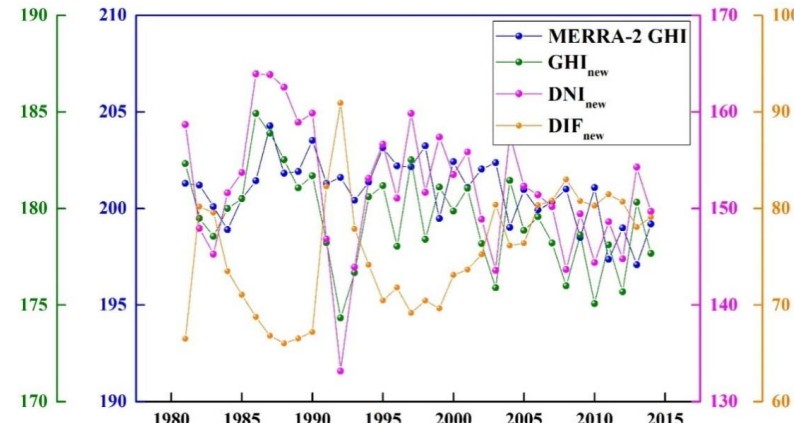

Figure 9. The annually variations of GHI, DNI and DIF throughout China. (The units for GHI, DNI and
DIF areWm⁻².)

Figure 9 indicates the yearly variations in GHI, DNI and DIF values over mainland China during

1981-2014. To better characterize the yearly variations in solar radiation in China, the anomaly GHI,
DNI and DIF values are calculated. Figure 10 illustrated the annual variations of the anomaly GHI, DNI
and DIF values in China. The results show that the MERRA-2 GHI values obviously overestimate the
GHI values in China during 1981-2014. Combined with the validation results of MERRA-2 GHI and
GHInew values, we think GHInew estimations fit the measured GHI values better. It is clear from Figure
9 and Figure 10 that GHI values have been gradually decreased from 1981 to 2014. The lowest annual
mean GHI (174.329 $Wm^{-2}$) and anomaly GHI (-50.914 $Wm^{-2}$) value occurred in 1992, which was
supposed to be caused by the strong aerosol radiative effect of volcanic eruption events in the Philippines
in 1992. The DNI values are directly proportion to GHI values with similar temporal variations, because
DNI is the main component of GHI values. The highest and lowest GHI values are found in 1981
(158.657 $Wm^{-2}$) and 1992 (133.137 $Wm^{-2}$), respectively. In contrast, DIF values show an opposite
temporal variation with GHI and DNI values. The DIF values have been gradually increasing from 1981
to 2014, especially in 1992 (DIF = 90.914 $Wm^{-2}$, anomaly DIF value = 15.544 $Wm^{-2}$) with an explosive
growth of DIF values. It is thought that DIF values are directly proportional to AOD value.

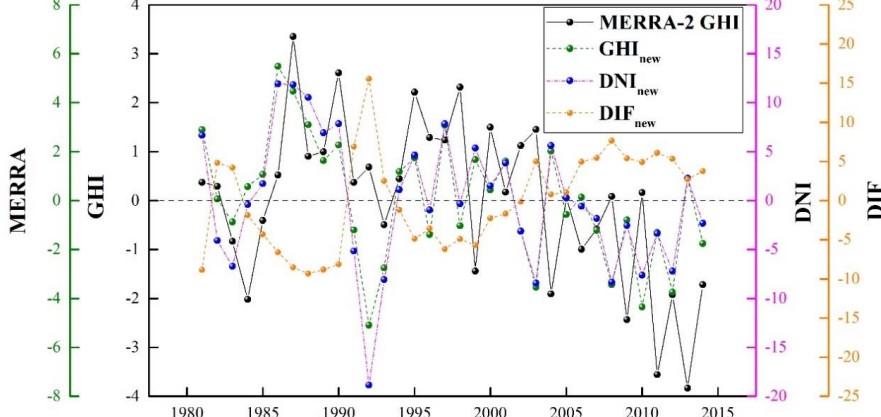

Figure 10. The annually variations of anomaly GHI, DNI and DIF throughout China (The units for
anomaly GHI, DNI and DIF are $Wm^{-2}$).
**4 Discussion**

The model accuracy of DNInew and DIFnew is relatively lower than that of GHInew, which may

be caused by three factors. First, there are too many low DNI and DIF values, because low DNI values
generally correspond to cloudy sky conditions, which may cause large uncertainties in DNI estimations.
Second, although they were the only available sunshine duration datasets with the highest density in





China, the point density of sunshine duration measurements at 2474 CMA stations was still sparse.
Finally, the spatial resolution of the input parameters derived from MERRA-2 products and the output
values (DNInew and DIFnew) were 0.50° (lat) *0.625° (lon), which may also degrade the accuracy of
the estimated DNI and DIF values.
Table 3. Validation of GHI estimations in China in previous studies.

| Parameters | Models/ Products | Format | Spatial resolution | Spatio-temporal continuity | RMSE (Wm$^{-2}$) | R |
|---|---|---|---|---|---|---|
| GHI | EPP | Grid | $1° \times 1°$ | Few vacancies | 34.028 | 0.930 |
| GHI | ISCCP-FD | Grid | $1° \times 1°$ | High continuity | 35.764 | 0.910 |
| GHI | GEWEX-SRB | Grid | $1° \times 1°$ | High continuity | 34.144 | 0.930 |
| GHI | EPP-TANG | Grid | 5km | Few vacancies | 33.912 | 0.930 |
| GHI | GLASS | Grid | $0.05° \times 0.05°$ | High continuity | 34.144 | 0.930 |
| GHI | MERRA | Grid | $0.625° \times 0.5°$ | High continuity | 85.775 | 0.822 |
| GHI | GHInew | Grid | $0.625° \times 0.5°$ | High continuity | 25.509 | 0.955 |

The estimated GHI values in this study were compared with other estimation results in previous
studies. The validation of DNI and DIF estimations in previous studies is not listed and discussed,
because the evaluation of DNI and DIF values over mainland China is not founded in previous studies.
Table 3 shows the validation results of GHI estimations in China in this study and previous studies.
Detailed descriptions of these models and products have been described in the Introduction Section. It
was clear that the surface GHI estimation results in this study show higher agreement with surface solar
radiation measurements at CMA stations than those of other estimation results in previous studies, which
may be due to the consideration of cloud effects on GHI. Although the spatial resolution of GHI
estimations by EPP-TANG is higher than GHInew, high spatiotemporal continuity and long temporal
ranges of the GHInew estimation results could remedy the defect of relatively lower spatial resolutions.

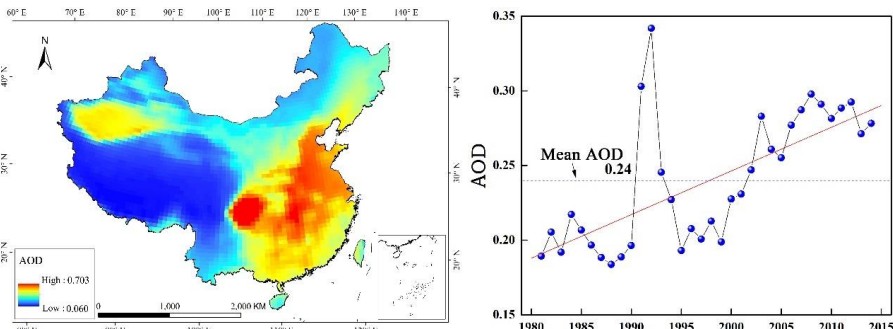

Figure 11. The spatial and temporal variations of AOD values during 1981-2014 throughout mainland
China.
It was supposed that the DIF values were strongly correlated to aerosol optical depth (AOD) in
China. Thus, we introduced the MERRA-2 AOD product to analyze the correlation between DIF values
and AOD values in China. Figure 11 show the spatial distribution of AOD values over mainland China.
As seen from Figure 8, Figure 9 and Figure 11, DIF values show similar spatial distribution patterns with
AOD values in China. The correlation coefficient between the annual mean DIF and annual mean AOD
values is 0.890. The Sichuan Basin is the area with the highest AOD (0.703) and the highest DIF (115.706
$Wm^{-2}$) values, while the Qinghai Tibetan Plateau is the area with the lowest AOD (0.060) and the lowest
DIF (42.928 $Wm^{-2}$) values. It is certain that AOD is an import factor in the DIF variations in China.
**5 Data availability**
The MERRA-2 Reanalysis data are available at GES DISC by NASA
(https://disc.gsfc.nasa.gov/daac-bin/FTPSubset2.pl). We provide these solar irradiance data in publicly
available repository: https://doi.org/10.6084/m9.figshare.10026563 (Qin, W. et al., 2019). The
corresponding author can be contacted for access meteorological data at CMA stations and solar
irradiance dataset generated in this study during 1981-2014 as well as ancillary data.
**6 Summary**
The applicability of REST2 in modeling GHI, DNI and DIF values using MERRA-2 reanalysis
products (*AOD550*, *p*, *rog* and *w*) and sunshine duration measurements at 2474 CMA stations throughout
China was tested in this study. Long-term grid GHI, DNI and DIF datasets (0.50° (lat) *0.625° (lon))
throughout China were then constructed. Finally, the spatiotemporal characteristics of GHI, DNI and DIF



in China were investigated.
The estimated SI values show high agreements with SI measurements at 17 CMA stations with
radiation measurements. Eighteen indicators including RMSE, MAE, RMSD, MAD, and MBD were
used to represent the model accuracy of the MERRA-2 GHI, GHInew, DNInew and DIFnew values. The
RMSE for MERRA-2 GHI, GHInew, DNInew and DIFnew are 85.775, 25.509, 46.852 and 35.700 Wm$^-$
$^2$, respectively; the MAE are 71.701, 18.993, 32.917 and 25.870 Wm$^{-2}$, respectively; and the R are 0.822,
0.955, 0.914 and 0.691, respectively. It could be concluded that the accuracy of MERRA-2 GHI values
has been significantly improved in this study. Relatively large estimation errors for MERRA-2 GHI and
GHInew values are found CHD in Sichuan Basin with GPI scores of 1.911 and -10.329, respectively,
because of the cloud air conditions there.
The spatiotemporal characteristics of GHI, DNI and DIF values from 1981-2014 over mainland
China were discussed using the generated grid GHI, DNI and DIF datasets in this study. The results show
that the MERRA-2 GHI records may overestimate the GHI values over mainland China. Generally, the
GHI and DNI values have gradually decreased from 1981-2014. However, DIF values have gradually
increased from 1981 to 2014, especially in 1992 (DIF = 90.914 Wm$^{-2}$, anomaly DIF value = 15.544 Wm$^-$
$^2$), which may be caused by the increasingly strong aerosol radiative forcing effects throughout China
during 1981-2014. The Qinghai Tibetan Plateau has always been the area with the highest GHI, highest
DNI and lowest DIF values (clear sky condition), while the Sichuan Basin has always been the area with
the lowest GHI, lowest DNI and highest DIF values (cloudy and rainy sky condition). It was validated
that the DIF values are strongly correlated with aerosol optical depth (AOD) in China.
Certainly, the REST2 model should be further validated in other climate zones around the world. As
discussed above, the GHI, DNI and DIF estimations are subject to input data quality, the interpolated
method and the relatively coarse resolution of MERRA-2 products. Further work should be conducted
to improve the accuracy of the GHI, DNI and DIF datasets generated in this study. Moreover, significant
relations between DIF and the AOD values are validated in this study, and further studies should be
undertaken to reveal the main driving factors for the spatio-temporal variations in GHI, DNI and DIF
values.
**Acknowledgements**
The   surface solar radiation (GHI, DNI and DIF values) data records at 17 sites of China were



obtained from National Meteorological Information Center of China http://data.cma.cn; MERRA-2
(Modern-Era Retrospective Analysis for Research and Applications, version 2) products were available
from Goddard Space Flight Center Level 2 and Atmosphere Archive Distribution System
http://ladsweb.nascom.nasa.gov. This work was financially supported by the National Natural Science
Foundation of China (No.41601044), the Special Fund for Basic Scientific Research of Central Colleges,
China University of Geosciences, Wuhan (CUGCJ1704, 007-G1323519253 and 111-162301182738),
and the 111 Project (grant No. B08030). We would like to thank Professor Christian A. Gueymard for
providing the code of the REST2 software. We would like to thank the China Meteorological
Administration (CMA) for providing the meteorological and radiation data.

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
