# Peer review of "A high-quality hourly, daily and monthly solar irradiance"

_Earth System Science Data, 2019_

## Referee Comment (RC1) · Anonymous Referee #1 · 10 Dec 2019

<A high-quality 1 hourly, daily and monthly solar irradiance dataset in China during 1981-2014 based on MERRA-2 Reanalysis products>

In this paper, the authors generated a high-quality hourly, daily and monthly solar irradiance dataset in China from 1981 to 2014 based on MERRA-2 data, and shown a high accuracy of estimated GHI, DNI and DIF compared with the ground-based measurements. Overall, this manuscript is clear and well written. However, the following minor questions are not satisfactorily answered:

(1) L85-86, duplicated word "Wei et al. Wei et al. (2019)". (2) L90-91, "Qin et al. (2015) developed an efficient physical parameterization (EPP) for estimating GHI values using MODIS land and atmospheric products and evaluated the EPP model at 91 CMA stations in China. However, the spatial resolution (1°x1°) and spatial continuity of the estimation results by the EPP model could not meet the requirements of solar energy research, which requires SI records with high spatial resolution". You mean the spatial resolution of Qin et al. 2015 is 1°x1°? (3) L138. How do you control of quality of CMA measurements? (4) What's mean of the MBD in eq. (12)? (5) There are too many indicators (eq. 4-20) for accuracy evaluation, the authors should write some brief introduction for these indicators. (6) L226. "Xianghe stations" means Xianghe have more than one stations? (7) Fig. 3. How many points used for evaluation of estimated hourly and daily GHI, DNI and DFI? (8) Why there are few strange points in the Fig. 8? (9) What's the MERRA official algorithm for estimating GHI? And why they daily GHI results with very high error (RMSE=85.78Wm-2) in Fig. 3.

―――――――――――――――――――――

---

## Referee Comment (RC2) · Anonymous Referee #2 · 17 Dec 2019

General comments This paper calculated a data set of solar irradiance based on MERRA-2 reanalysis data and sunshine duration data over China. The description of method, data and the comparisons with other studies is ambiguous and make the result look dubious. Using daily cumulative sunshine duration to derive hourly cloud transmittance and hourly solar irradiance is illogical and maybe an obvious mistake. Also, I don't believe the accuracy of satellite retrievals would lower than that of reanalysis data corrected with ground observations (except for the station used to correct the reanalysis) because the reanalysis data is difficult to simulate the realistic clouds. I personally think the paper is outside the scope of regular articles because it is similar to the interpretation of data.

[Figure]

Specific comments 1. The words "high-quality" in title is inappropriate because its spatial resolution is coarse. 2.Most of the error indicators for accuracy evaluation are not frequently used and also redundant. 3. Data description is unclear. For example, the observation instruments, error, frequency, length, quality and so on. Moreover. The quality control method should be introduced in detail because it is very significant for accuracy evaluation. 4. How about the uncertainty about the interpretation of sunshine duration data because it is a main factor influencing the accuracy of your products. 5. how to derive the cloud transmittances, and how about its uncertainty? how to correct the solar irradiance of MERRA-2 with the cloud transmittances? How about the uncertainty of the correction process? 6. How to derive the hourly cloud transmittance because the sunshine duration is daily cumulative. This is the obvious mistake of this article. 7. In Fig.3 for GHInew, we can not observe the overall overestimation, but the MAE indicates that the GHInew is significantly overestimated. It's a contradiction and the results is unbelievable. 8. Table 3 is meaningless and the comparison with other studies is extremely unfair because the spatiotemporal resolution, the input data, and the observation data (also number of observation stations) for these studies are completely different with you.

---

## Author Comment (AC1) · 17 Dec 2019

Dear Referee #1: Thank you very much for your positive and constructive comments. We have studied the valuable comments carefully and have made corrections which we hope can meet with approval. The point to point responds to the reviewer's comments are listed as following:

Comment#1: (1) L85-86, duplicated word "Wei et al. Wei et al. (2019)"..

[Response to Comment#1]: Thank you for your valuable advice. We have changed "Wei et al. Wei et al. (2019)" to "Wei et al. (2019)" in the revised manuscript.

[Figure]

Comment#2: L90-91, "Qin et al.(2015) developed an efficient physical parameterization (EPP) for estimating GHI values using MODIS land and atmospheric products and evaluated the EPP model at 91 CMA stations in China. However, the spatial resolution (1âŮęx1âŮę) and spatial continuity of the estimation results by the EPP model could not meet the requirements of solar energy research, which requires SI records with high spatial resolution". You mean the spatial resolution of Qin et al. 2015 is 1âŮęx1âŮę?

[Response to Comment#2]: Thank you very much for your valuable comments. We are deeply sorry for this careless mistake. We intended to express that the spatial continuity of the estimated results by EPP model could not meet the requirements of solar energy research, because there are many vacancies in MODIS Level2 land and atmosphere products. We have changed "the spatial resolution (1° 1°) and spatial continuity...with high spatial resolution" to "the spatial continuity...with high spatial continuity" in L90-91 in the revised manuscript.

Comment#3: L138. How do you control of quality of CMA measurements?.

[Response to Comment#3]: Thank you very much for your valuable suggestions for this manuscript. These CMA measurements used in this study have been checked manually using extreme value test and time consistency test method by China Meterological Administration (CMA). The quality control processes for solar irradiance measurements at CMA stations have also been conducted in this manuscript as follows: each measured GHI, DNI and DIF values should not exceed the solar irradiance at the top of the atmosphere at the same geographical location. Detail steps for controlling the data quality of solar irradiance measurements have been added in the revised manuscript.

Comment#4: What's mean of the MBD in eq. (12)?.

[Response to Comment#4]: Thank you very much for your valuable suggestions for this manuscript. We are sorry for this mistake. It's MAD (the relatively mean absolute bias error) not MBD. We have corrected this error in the revised manuscript.

[Figure]

Comment#5: There are too many indicators (eq. 4-20) for accuracy evaluation, the authors should write some brief introduction for these indicators.

[Response to Comment#5]: Thank you very much for your nice comment. We have briefly introduced these indicators in the revised manuscript. Meanwhile, we also paste the reference (Gueymard 2014) that described these indicators in detail in the revised manuscript. We are apologized that we could not describe the definition for each indicator, limited by the space of the article. We are sorry again for that. However, if you think this is a very important issue, we will write detail introduction of each indicator in the next round review.

Comment#6: L226. "Xianghe stations" means Xianghe have more than one stations.

[Response to Comment#6]: Thank you very much for your valuable suggestions for this manuscript. "Xianghe stations" have been changed into "Xianghe station" in the revised manuscript.

Comment#7: Fig. 3. How many points used for evaluation of estimated hourly and daily GHI, DNI and DFI?.

[Response to Comment#7]: Thank you very much for your comments. The number of the data sample for evaluating the estimated daily GHI, DNI and DIF is 129662. The number of the data sample for evaluating the estimated hourly GHI, DNI and DIF at Wuhan station and Xianghe station are 10032, 25819, respectively. The number of the data sample for evaluating the estimated hourly GHI at CERN stations is 3857998. The number of data samples have been described in the revised manuscript.

Comment#8: Why there are few strange points in the Fig. 8?.

[Response to Comment#8]: Thank you very much for your kind reminder. We are sorry for this mistake. There was a software failure when plotting the map. We have added a new corrected map in the revised manuscript.

Comment#9: What's the MERRA official algorithm for estimating GHI? And why they

daily GHI results with very high error (RMSE=85.78Wm-2) in Fig. 3.

[Response to Comment#9]: Thank you for your nice comment on this manuscript. The MERRA-2 is produced with version 5.1.4 of the GEOS atmospheric data assimilation system. The key components of the system are the GEOS atmospheric model and the GSI analysis scheme. The Radiative transfer calculations necessary for the assimilation of satellite radiances in MERRA-2 are performed using the CRTM. Detail information for the official algorithm for calculating GHI could be found in an article named "The Modern-Era Retrospective Analysis for Research and Applications, Version 2 (MERRA-2)". The very high error (RMSE=85.78Wm-2) for MERRA-2 GHI records in Fig. 3 is a combined result of multiple factors. The accuracy of the cloud effect on GHI may be the main reason, since cloud was considered to be the most uncertain factor for predicting surface solar irradiance, owing to the variation of the cloud shape, cloud type and cloud phase in various climatic zones and terrain features. The high errors of the GHI values derived from MERRA-2 are consistent with a known tendency for the GEOS-5 systems to underestimate mid-latitude continental cloud cover (Draper et al. 2018). Further studies should be conducted in future studies to found the limitation of accuracy of GHI records in MERRA-2. This explanation for the high errors of MERRA GHI records have been added in the revised manuscript.

Please also note the supplement to this comment:
https://www.earth-syst-sci-data-discuss.net/essd-2019-204/essd-2019-204-AC1-supplement.pdf
* * *

---

## Author Comment (AC2) · 4 Jan 2020

Dear Referee #2:

Thank you very much for your positive and constructive comments. We have studied the valuable comments carefully and have made corrections which we hope can meet with approval. We apologize that we did not present some important results and Figures in the original manuscript, limited by the length of the article. We will add these Figures and results in the revised manuscript or supplemental material, following the suggestions by Referee #2 and the editor. The point to point responds to the Referee #2's comments are listed as following:

**Comment#1:**

Using daily cumulative sunshine duration to derive hourly cloud transmittance and hourly solar irradiance is illogical and maybe an obvious mistake.

**[Response to Comment#1]:**

Thank you for your valuable advice. The actual sunshine duration during a given period is defined as the sum of that sub-period for which the direct solar irradiance exceeds 120 W  $m^{-2}$  (WMO). Sunshine duration is a potential good index that can reflect the influence of cloud on surface solar irradiance. The sunshine duration data are widely used to correct the cloud effect on global (direct and diffuse) radiation. The REST2 could efficiently estimate the GHI, DNI and DIF values, but should be corrected under all sky condition. However, getting hourly sunshine durations measurement throughout mainland China with long temporal range is impossible. Therefore, we tested the accuracy of correcting the solar irradiance values using daily sunshine duration measurements at Xianghe station. Figure 1 showed the line chart of the GHI estimations in clear sky, GHI estimations with cloud correction, GHI observation in random selected day of each month in 2007 at Xianghe station. Figure 2 showed the comparison results of the hourly estimated GHI/DNI/DIF values without cloud correction and the hourly measured GHI/DNI/DIF values at Xianghe station. The result showed that there are uncertainties in estimating GHI/DNI/DIF values without considering the cloud effect on solar irradiance. The accuracy of the estimated hourly GHI, DNI and DIF

values have been significantly improved after correcting the cloud effect on GHI, DNI and DIF values.

Figure 1. The hourly estimated GHIclr, GHIcc and GHIm in random selected day of each month in 2007 at Xianghe station. (GHIclr, GHIcc, and GHIm are the GHI estimations in clear sky, GHI estimations with cloud correction, GHI observation, respectively).